# Characterization of a hemolytic and antibiotic-resistant *Pseudomonas aeruginosa* strain S3 pathogenic to fish isolated from Mahananda River in India

**Dipanwita Ghosh[1], Preeti Mangar[2], Abhinandan Choudhury[1], Anoop Kumar[1], Aniruddha Saha[2], Protip Basu[3], Dipanwita Saha[1]***

**1** Department of Biotechnology, University of North Bengal, Siliguri, West Bengal, India, **2** Department of Botany, University of North Bengal, Siliguri, West Bengal, India, **3** Department of Botany, Siliguri College, West Bengal, India

* dipanwitasaha@nbu.ac.in

## Abstract

Virulent strain *Pseudomonas aeruginosa* isolated from Mahananda River exhibited the highest hemolytic activity and virulence factors and was pathogenic to fish as clinical signs of hemorrhagic spots, loss of scales, and fin erosions were found. S3 was cytotoxic to the human liver cell line (WRL-68) in the trypan blue dye exclusion assay. Genotype characterization using whole genome analysis showed that S3 was similar to *P. aeruginosa* PAO1. The draft genome sequence had an estimated length of 62,69,783 bp, a GC content of 66.3%, and contained 5916 coding sequences. Eight genes across the genome were predicted to be related to hemolysin action. Antibiotic resistance genes such as class C and class D beta-lactamases, *fosA*, *APH*, and *catB* were detected, along with the strong presence of multiple efflux system genes. This study shows that river water is contaminated by pathogenic *P. aeruginosa* harboring an array of virulence and antibiotic resistance genes which warrants periodic monitoring to prevent disease outbreaks.

## Introduction

One of the most important natural resources is water which is essential in the life of all living organisms. However, excess runoff of sewage and fertilizers into freshwater bodies provides a favorable environment for the growth and reproduction of pathogenic microorganisms [1]. It is reported that river water is usually contaminated by diverse groups of pathogenic bacteria [2]. *Pseudomonas aeruginosa* is a Gram-negative, rod-shaped and uniflagellated bacterium that can colonize a wide range of habitats including soil, water, animals, plants, and humans [3]. It is a well-known pathogen for opportunistic infections, which causes morbidness and even death in immunocompromised and cystic fibrosis-affected patients [4,5]. As a threatening fish pathogen, *P. aeruginosa* induces ulcerative syndrome and hemorrhagic septicemia leading to high mortality and significant economic losses [6]. Since water is the natural reservoir of this

**Data Availability Statement:** All relevant data are within the manuscript and its Supporting Information files.

**Funding:** The author(s) received no specific funding for this work.

**Competing interests:** The authors have declared that no competing interests exist.

bacterium, the presence of *P. aeruginosa* is considered an indicator of contamination in water by some researchers [7,8]. This species is present in high concentrations in domestic wastewater and is a major contaminant in surface runoff. However, due to the low fecal abundance of this bacterium, agricultural leachates and barn effluents are also thought to be other important sources of *P. aeuginosa* [9]. *P. aeruginosa* displays a wide range of virulence systems which contributes to its ability to cause infection and colonization [10]. Mechanisms of virulence in these bacteria include both cell-associated factors such as lipopolysaccharides, flagella, pili, an array of extracellular secretions such as proteases, exotoxins, elastases, siderophores, extracellular polysaccharides [10,11]. Hemolysins are extracellular toxic proteins, produced by many Gram-positive and Gram-negative bacteria with lytic activities [12]. Also called cytolysins, they are responsible for forming pores in membranes of various host cells including red blood cells (RBCs), epithelial cells, and leukocytes leading to cell lysis and ultimately cell death [13,14]. The investigations on hemolysins of *P. aeruginosa* are mainly based on clinical isolates [15–17] although the hemolytic trait is reported to occur in environmental isolates as much as in clinical isolates [18]. A study reported that hemolytic *P. aeruginosa* PAO1 causes plant damage when present in the rhizosphere and disruption of the hemolysin gene lowers virulence towards poplar and barley [19].

High incidences of multidrug resistance in Gram-negative pathogenic bacteria have made the treating of bacterial diseases extremely challenging. *P. aeruginosa* readily achieves resistance to various classes of antibiotics through intrinsic and adaptive mechanisms that include low permeability of the bacterial outer membrane, biofilm formation, and involvement of robust efflux pumps, which are naturally present in bacteria to combat toxic materials in the environment [20]. In addition, acquiring resistance through horizontal gene transfer and mutational resistance has led to it joining the ranks of superbugs [21].

The current study aimed to investigate the range of virulence and antibiotic resistance in pathogenic bacteria present in river waters of North Bengal, India. Preliminary studies led to the isolation of a virulent strain *Pseudomonas aeruginosa* S3. The strain was further studied for its susceptibility to 21 common antibiotics belonging to various classes. Pathogenicity of S3 was tested in *Anabus testudineus* fishes and cytotoxicity was tested in the human liver cell line. In addition, whole genome sequencing of S3 provided information to predict genes related to virulence factors and antimicrobial resistance along with those involved in hemolytic activity.

## Materials and methods

### Isolation and screening of putative hemolytic strains based on hemolytic activity

The River Mahananda is a major river in North India and has a vital role in regulating the economy of the adjoining areas [22]. The river flows through the Himalayan mountains where it originates and descends into the plains in the sub-Himalayan region of Darjeeling district in West Bengal flowing through the Mahananda Wildlife Sanctuary. Then it flows through the heart of Siliguri city and meanders along several places before merging with the River Ganga. Water samples were withdrawn in sterilized vials from different locations along the Mahananda River in Siliguri.

The vials with sampled river water were kept at 4°C and studied within 24 h of sample collection. To isolate hemolytic bacteria, serial ten-fold dilutions of the samples were made in sterile distilled water. Then, 0.1 mL of each dilution was spread plated on tryptone soya agar (TSA) medium (Himedia Laboratories, Mumbai, India) supplemented with 5% (v/v) sheep blood. The plates were incubated at 37°C for 24h following which they were observed against the light for halo formation around the bacterial colonies. Altogether 26 colonies were picked

up based on positive halo formation from the plates and axenic cultures were maintained by streaking them on nutrient agar slants [23]. For long time preservation, the cultures were stored in glycerol stock (10% v/v glycerol) and kept at -20°C.

## Hemolytic activity

The hemolytic activity of the bacterial isolates was determined as described previously [24] with minor modifications. Briefly, sheep red blood cells were prepared by washing thrice with phosphate-buffered saline (PBS). Culture supernatant of bacterial isolates was collected by centrifuging 24 h old culture of each bacterium in tryptone soya broth(TSB). For assaying hemolysis, 245μl of culture supernatant was mixed with 5μl of washed sheep RBC and 10mM $CaCl_2$ and incubated for 1h at 37°C. Positive control contained 1% Triton X-100 in place of culture supernatant while sterile TSB was used for negative control. After incubation, samples were centrifuged and hemoglobin release was monitored by estimating the optical density at 540nm ($OD_{540}$). For normalization, the negative control reading was subtracted from each sample reading. Percent hemolysis was calculated by setting the Triton X-100 control as 100% hemolysis. All the experiments were performed in triplicate and the data were represented as triplicate of mean ± SD. To find the significant differences ($P<0.05$) between the hemolytic isolates, hemolysin assay experiments were analyzed by one-way ANOVA in SPSS Software version 21.

## Detection of phenotypic attributes of virulence

Production of protease was determined in skimmed milk supplemented (1.5%w/v) nutrient agar medium. This medium was inoculated by each isolate and incubated for 24h at 37°C. Bacteria that were able to degrade casein exhibited a zone of clearance around the growth of bacteria [25]. DNase production by the isolates was observed in DNase agar medium (HiMedia laboratories, Mumbai) where the media plates were inoculated with the strains and incubated for 24h at 37°C. Bacterial colonies surrounded by pale pink to white halos were identified as DNase positive [26]. For testing the production of lipase, the isolates were allowed to grow in Tween-80 medium at 37°C for 24 h. The clear zone surrounding the streaked bacterial cultures due to the crystal formation indicates lipase production [27]. Amylase production by the isolates was tested in starch agar plates according to the method of Barrow and Feltham. Production of siderophore was examined by the Chromazurol S (CAS) assay [28].

## Biochemical characterization

The isolate showing high hemolytic activity, multiple virulence traits, and antibiotic resistance was subjected to further investigations. The selected strain S3 was characterized through biochemical tests which included Gram staining, catalase, oxidase, indole, methyl red, Voges Proskauer, oxidation-fermentation, nitrate reduction and citrate utilization [29]. Pigment production was tested in Pseudomonas agar medium (Himedia Laboratories, India).

## Antimicrobial susceptibility tests

To determine the antimicrobial sensitivity of *Pseudomonas aeruginosa* S3, the Kirby–Bauer disk diffusion method was used as per the standards of CLSI and EUCAST [30]. Initially, S3 was grown in nutrient broth for 24 h at 37°C. Sterile Mueller–Hinton agar (MHA) plates were spread with S3 culture using a cotton swab and the inoculated plates were dried at room temperature. Subsequently, antimicrobial discs (Hi-media Laboratories, Mumbai, India) were placed on the inoculated plates and incubated at 37°C for 24 h. The antimicrobial substances

used in this study were ampicillin (10 mcg), amoxicillin/clavulanic acid (10 mcg), penicillin (10 mcg), cefoperazone (75mcg), ciprofloxacin (5mcg), gentamicin (10mcg), streptomycin (25 mcg), kanamycin (30 mcg), norfloxacin (10mcg), ofloxacin (5mcg), imipenem (10mcg), cefepime (30mcg), cefixime (5mcg), cefotaxime (30mcg), cefuroxime (30 mcg), chloramphenicol (30 mcg), co-trimoxazole (15 mcg), trimethoprim (5 mcg), tetracycline (30 mcg), oxytetracycline (30 mcg) and colistin (10 mcg). Colistin susceptibility was tested by the disk elution method following CLSI guidelines. For this, specific number of colistin disks were added to 10 ml sterile cation-activated Mueller–Hinton broth (0 disk = control, 1 disk = 1 μg/mL, 2 disks = 2 μg/mL and 4 disks = 4 μg/mL) and vortexed gently for 40 min for the elution of colistin into the medium. For inoculum preparation, 3 to 5 isolated pure colonies of S3 from overnight nutrient agar plates were suspended in normal saline and added to each test-tube to attain a final inoculum density of $7.5 \times 10^5$ cfu/mL. The inoculated tubes were again vortexed at low speed and incubated overnight at 35°C. Categorization of the strain S3 as sensitive (S) or resistant (R) towards each antimicrobial substance was done according to the suggestion mentioned in the CLSI and EUCAST interpretative guidelines for *Pseudomonas. Pseudomonas aeruginosa* ATCC 27853 was used as a reference strain for the antimicrobial sensitivity experiments.

## Cell cytotoxicity assay

Human liver embryonic cell line WRL-68 was purchased from NCCS (National Centre for Cell Science) in Pune, Maharashtra, India, and used to check the cytotoxicity of the strain *Pseudomonas aeruginosa* S3. The strain S3 was inoculated in the Luria-Bertani (LB) broth and incubated for 24 h at 37°C with an agitation speed of 90 rpm. Cell-free supernatant (CFS) was collected by centrifuging the 24 h culture at 10,000g for 30 mins at 4°C followed by filtering through a 0.45 μm membrane filter. Culture filtrate of *Lactobacillus* sp., a non-pathogenic strain, was also prepared following the same procedure and included as the positive control in the experiment. WRL-68 cells were allowed to grow in Dulbeco's modified Eagle's medium (DMEM) with 10% foetal calf serum in an atmosphere containing 5% $CO_2$. For experimental purpose, cells were added to culture plates (60 mm) and incubated in $CO_2$ incubator for 24 h at 37°C. Subsequently, the cell free supernatants (1.5 mL) were added to the cells in separate sets and incubated at 25°C for one hour. Sterile LB medium was added to the control set. Following incubation, the cells were observed under phase-contrast inverted microscope (Olympus CK40-SLP) at 200X magnification. The percentage of viable cells of treated and control sets was determined following the trypan blue dye exclusion assay described previously [31]. Percent viability was determined as: [total number of viable cells per ml of aliquot/ total number of cells per ml of aliquot] × 100. The experiment was repeated thrice and the mean was computed. Standard error was calculated using the statistical software OriginPro R version 9.9 freely available from https://www.originlab.com.

## Pathogenicity testing in fishes

Healthy small-sized fishes (25–30 g) of *A, testudineus* were used for testing the pathogenicity of S3 in fish. In this current study, fishes were procured from local fisheries in the Darjeeling district and acclimatized for 15 days in glass aquaria (90X35X35 cm). Each aquarium contained 8 fish and the water temperature was maintained between 25–30°C. For pathogen injection, S3 was grown in TSB for 18 h at 37°C under an agitation speed of 90 rpm and subsequently centrifuged to collect the cells which were resuspended in 0.85% saline solution to obtain a cell density of $1 \times 10^7$ cfu/mL [32]. Benzocaine at a concentration of 25 mg/L was used to anesthetize the fishes where the fishes were kept in the solution for 1–2 min. The S3

suspension was injected into each of the fish at a concentration of 0.4 mL/25 gm body weight and maintained in a separate aquarium [30]. The fishes injected with 0.85%w/v saline solution at a similar dose were marked as a control group. The pathological lesions and death of fish were monitored and recorded every 24 h post administration for one week. The approval (IAEC approval No. IAEC/NBU/2022/39) for conducting the above experiment was obtained from Institutional Animal Ethics Committee (IAEC), Department of Zoology, University of North Bengal under the guidelines of CPCSEA, New Delhi. The fishes were anesthetized by keeping in benzocaine solution (25mg/L) for 1-2min.

## Whole-genome sequencing, assembly, annotation, and comparative genome analysis

The selected strain S3 was inoculated into TSB and grown for 24 h at an incubation temperature of 37˚C. The chromosomal DNA of the bacterium was isolated using a bacterial gDNA isolation kit (Xploregen Discoveries, Bangalore, India) following the protocol obtained from the manufacturer. DNA concentration was determined using the Qubit double-stranded DNA (dsDNA) high-sensitivity (HS) assay kit (Invitrogen, USA) in a Qubit 3 Fluorometer. Total DNA was enzymatically digested to obtain the average fragment size ranging from 200–300 bp. and libraries were generated by following the NEBNext Ultra II protocol. Fragments were analyzed in Agilent 2100 bioanalyzer and sequencing was conducted with the Illumina HiSeq 4000 sequencer using the 2x150 paired end sequencing strategy to generate 15810782 reads. Read quality was assessed using FastQC and MultiQC and trimming was done using TrimGalore (version 0.6.4) to remove adapter sequences. Primary assembly of the filtered reads was done using Unicycler with default settings (version 0.4.8). MOB-suite (version 3.1.0) was used to predict plasmid sequences from the primary assemblies which showed negative results. The final finishing of the genome was done using Contiguator (https://contiguator.sourceforge.net/).

Annotation of the S3 genome was done using Rapid Annotations using Subsystems Technology (RAST) used at Pathosystems Resource Integration Center (PATRIC) and Prokka tools. The circular genome map and phylogenetic tree were created using PATRIC. Sequences for virulence factors were searched and analyzed through the Virulence Factor Database (VFDB) and Victors resource. The Comprehensive Antibiotic Resistant Database (CARD) and the NCBI National Database of Antibiotic Resistance Organism (NDARO) were used to detect antimicrobial resistance genes. In addition, hemolysin genes were predicted using the PATRIC annotation service and analyzed through BLASTP. The genomes or contigs were aligned against each other using the tool MAUVE version 20150226 for comparison [33].

## Phylogenetic analysis

Using the PubMLST server [34], the identification of *Pseudomonas aeruginosa* S3 was confirmed by rMLST (ribosomal multilocus sequence typing) [35]. The PATRIC Phylogenetic Tree Building Service [36] was used to create a phylogenetic tree using amino acid and nucleotide sequences from a selected number of the BV-BRC global Protein Families (PGFams) [37]. MUSCLE was used to align the protein sequences [38] whereas Biopython was used to align the coding gene nucleotide sequences [39]. The aligned protein and nucleotides were formatted into a PHYLIP file for ease of use, and a partition file was created specifically for RaxML analysis [40]. With the help of RaxML, 100 rounds of the "Rapid" bootstrapping option [41] support values were created. The phylogenetic tree was visualized by FigTree [42].

### Acquisition of the accession numbers for the nucleotide sequence

The draft genomes of S3 were submitted in the NCBI database under the accession number JAESVH000000000. The BioProject ID in GenBank is PRJNA693431. The organism was identified as *Pseudomonas aeruginosa* S3.

## Results

### Hemolytic activity and virulence-related traits of isolated strains

Altogether 26 bacterial strains were observed to be hemolytic among 256 strains isolated from the Mahananda River during primary screening. A quantitative estimation showed that strain S3 possessed the highest hemolytic activity (Fig 1). Further studies on virulence-related phenotypes of the isolates showed that S3 possessed the maximum number of virulence traits. It showed protease, DNase, and lipase activity and produced a siderophore in the CAS medium (S1 Table in S1 File).

### Biochemical characterization and antimicrobial resistance profile of S3

The strain S3 was observed to be Gram-negative and rod in shape which produced green fluorescent pigment on Pseudomonas agar medium. The strain showed positive results in catalase, oxidase, citrate utilization and nitrate reduction tests while it showed negative results for indole, methyl red, and Voges Proskauer tests. It was found to be oxidative in the oxidation-fermentation test. These results match the phenotype of *P. aeruginosa* described in Bergey's manual [43].

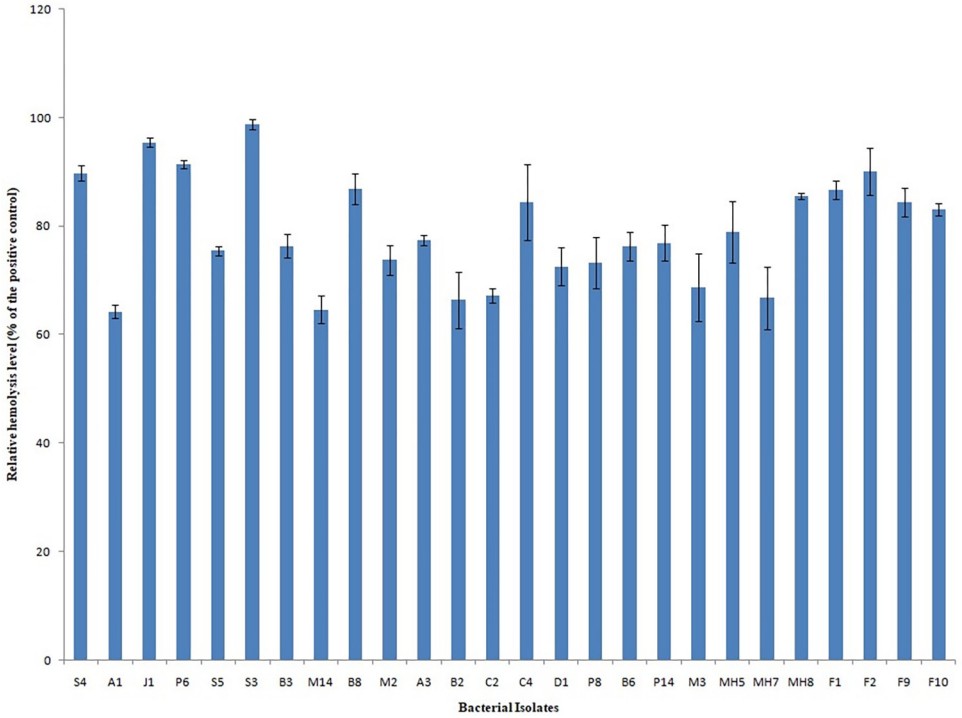

**Fig 1. Hemolytic activity of the bacterial isolates.** Data are presented as triplicate of mean ± SD. One-way ANOVA suggests a significant difference ($p < 0.05$) in hemolysis (%) by the isolates.

**Table 1. Antimicrobial sensitivity testing of *P. aeruginosa* S3.**

| Class | Antimicrobial substance | Disk content (mcg) | Interpretation |
|---|---|---|---|
| Aminoglycosides | Gentamicin | 10 | S |
| | Streptomycin | 25 | R |
| | Kanamycin | 30 | R |
| β-lactam | Penicillin<br>Ampicillin<br>Amoxycillin/Clavulanic acid | 10<br>10<br>10 | R<br>R<br>R |
| Carbapenems | Imipenem | 10 | R |
| Fluoroquinolones | Ciprofloxacin | 5 | S |
| | Ofloxacin | 5 | S |
| | Norfloxacin | 10 | R |
| Cephalosporin | Cefepime | 30 | R |
| | Cefixime | 5 | R |
| | Cefotaxime | 30 | R |
| | Cefoperazone | 75 | R |
| | Cefuroxime | 30 | R |
| Phenicols | Chloramphenicol | 30 | R |
| Folate pathway inhibitors | Co-trimoxazole | 15 | R |
| | Trimethoprim | 5 | R |
| Tetracyclines | Tetracycline | 30 | R |
| | Oxytetracycline | 30 | R |
| Polymyxins | Colistin | 10 | R |

The antibiotic resistance profile of *P. aeruginosa* S3 is listed in Table 1, which reveals that among the 21 antimicrobial substances tested, strain S3 was resistant to 19 of them. It showed resistance to all tested beta-lactam class of antibiotics including penicillins, all five cephalosporins, and carbapenem. S3 was also resistant to chloramphenicol, colistin, tetracyclines, and folate pathway inhibitors. It showed resistance to the aminoglycosides streptomycin and kanamycin but was susceptible to gentamycin. Similarly, it was found to be susceptible to fluoroquinolones, ciprofloxacin, and ofloxacin but was resistant to norfloxacin.

## Pathogenicity in fish

The pathogenicity of S3 was tested by injecting it into *A. testudineus* fishes intramuscularly. Almost 90% mortality was recorded within the first 24 h and by 36 h, all injected fishes were dead. The fishes developed clinical signs like hemorrhagic spots on the body surface especially on the head and ventral aspect of the abdomen, loss of scales, fin erosions, and exophthalmia (Fig 2).

## Cell cytotoxicity assay

The CFS of S3 showed cytotoxicity towards human liver cell line WRL-68. The trypan blue dye exclusion assay revealed a considerable decrease in cell viability when WRL-68 cells were exposed to the CFS of S3 compared to the controls (Fig 3). The viability of the S3-treated WRL-68 cell was observed to be 3.1% which was significantly lower compared to the cell treated with *Lactobacillus* (69%).

## Genetic features and phylogeny of S3

Genomic characteristics and annotated gene information of strain S3 are briefly presented in Table 2. The draft genome length of strain S3 was 62,13,354 bp with a GC composition of

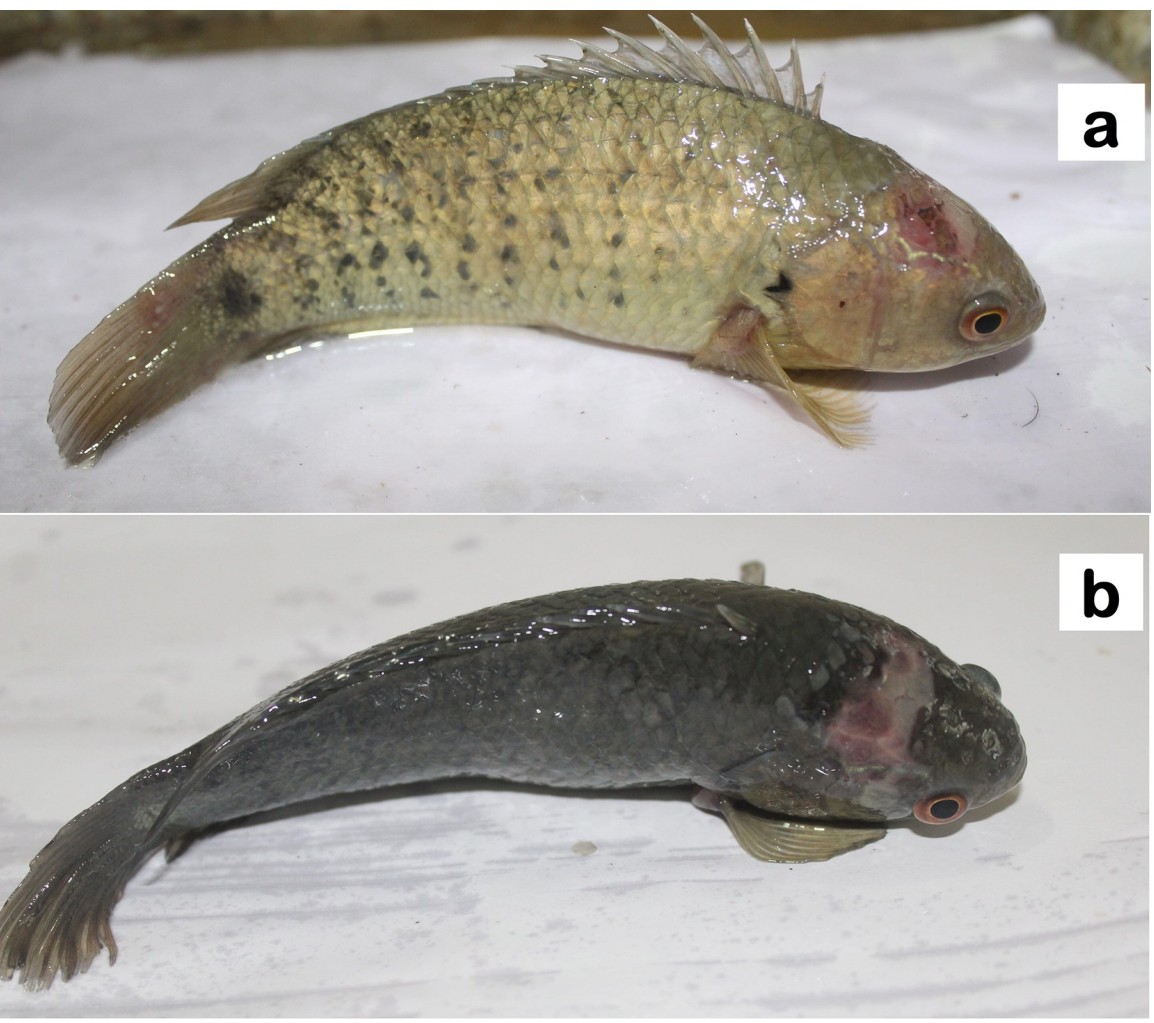

**Fig 2. Pathogenicity testing of *Pseudomonas aeruginosa* S3 in *Anabaena testudineus*.**

66.32% and harbored 5916 protein-coding sequences. The circular viewer application of PATRIC was used to construct the circular whole genome map of S3 and shown in Fig 4. Prospective genes obtained from the draft genome of S3 were analyzed and annotated using the subsystem approach of the RAST server (Fig 5). Subsystem refers to an assemblage of functionally related protein families, collectively referring to a precise biological process or structural complexes [44]. It had been observed that in strain S3, membrane transport function was controlled by 166 genes, biotic and abiotic stress response was regulated by 104 genes, virulence, defense, and disease-related function was controlled by 62 genes, and whereas metabolism and acquirement of iron were taken care of by 52 genes. Careful analysis of the genome of S3 also revealed the involvement of four genes associated with prophages, transposable elements, and plasmids. Listeria Pathogenicity Island LIPI-1 extended was found to be present under this subsystem. The subsystem's description obtained at the SEED viewer revealed the presence of *prf A*, *plcA*, *plcB*, *mpl*, and *actA* in LIPI-1 extended.

A Phylogenetic tree using the core genes of the whole genome sequence of S3 was constructed using the PATRIC database and presented in Fig 6. The microbial taxonomy of S3 was confirmed as *Pseudomonas aeruginosa* at 100% identity based on the variation of 56 genes

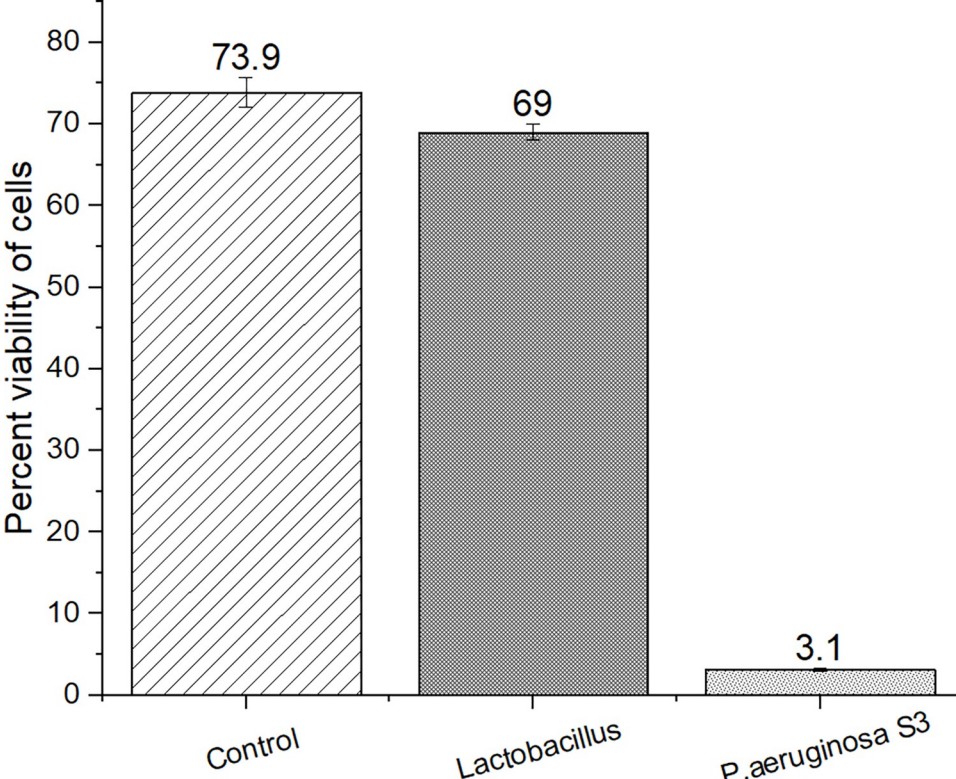

**Fig 3. Cytotoxic effect of *Pseudomonas aeruginosa* S3 cell-free culture supernatant fluid on human liver cell line (WRL-68).**

**Table 2. Genomic characteristics and annotation information of the chromosome of *Pseudomonas aeruginosa* S3 based on PATRIC.**

| Genome Features | Chromosome |
|---|---|
| Genome length (bp) | 62,69,783 |
| Protein-coding genes | 5916 |
| GC content (%) | 66.32298 |
| The number of tRNA | 55 |
| The number of rRNA | 3 |
| Contigs | 46 |
| Contig L50 | 1 |
| Contig N50 | 61,73,608 |
| Coarse Consistency(%) | 99.9 |
| Fine Consistency(%) | 99.7 |
| CheckM Completeness | 99 |
| CheckM Contamination | 0.6 |
| Hypothetical proteins | 1112 |
| Proteins with functional assignments | 4804 |
| Proteins with Subsystem assignments | 2073 |
| Proteins with EC number assignments | 1288 |
| Proteins with GO assignments | 1095 |
| Proteins with Pathway assignments | 971 |

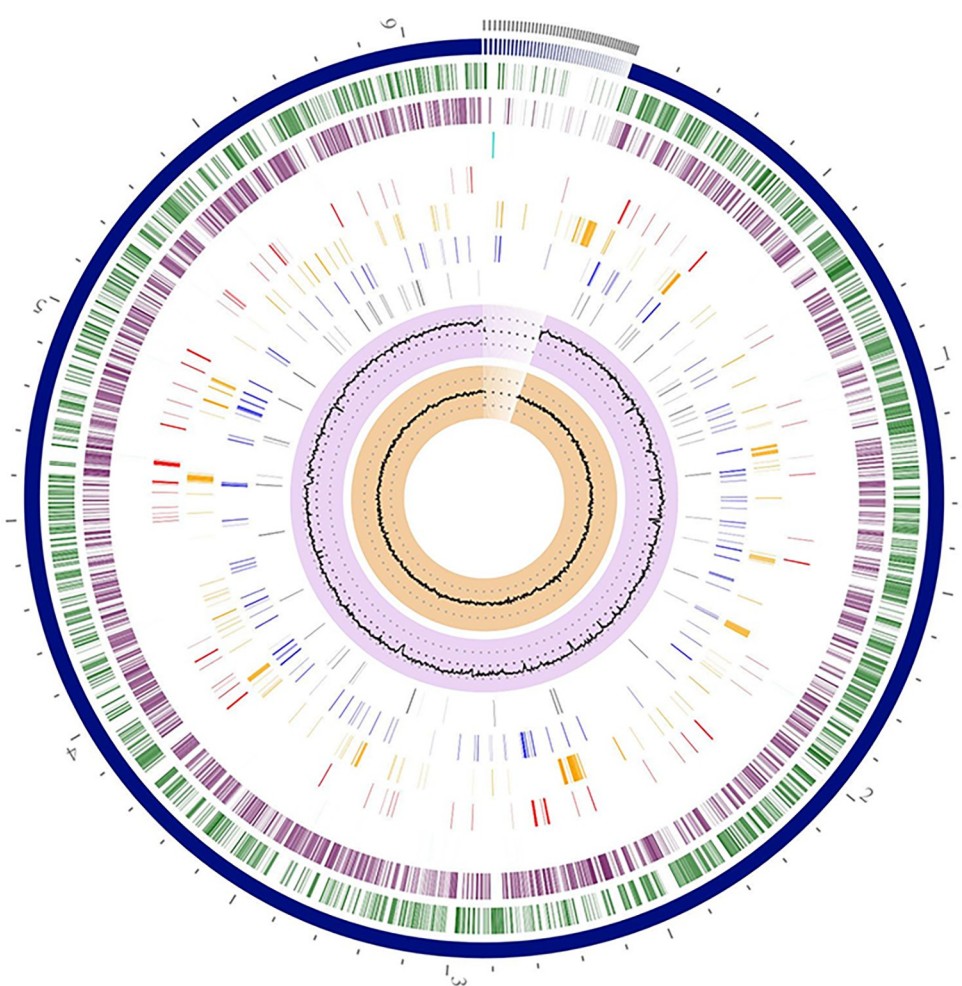

**Fig 4. The circular genome visualization of *P. aeruginosa* S3 was created using the PATRIC circular viewer with the layout starting from the outermost layer contigs and moving towards the center, with forward and reverse coding sequences, non-CDS features, AMR genes, VF Genes, transporters, and drug targets.** The two inner tracks are GC content and GC skew.

encoding ribosomal protein subunits. Pairwise alignment of S3 with whole genomes of *P. aeruginosa* PAO1 and *P. balearica* DSM 6083 was done using progressive MAUVE and S3 showed the highest similarity with *P. aeruginosa* PAO1 as shown in Fig 7. The sequence type was found as 549 using the MLST database which also matched with *P. aeruginosa* PAO1.

## Prediction of genes encoding virulence factors, antibiotic resistance, and hemolysin action

Several genes were predicted as virulence factors in the draft genome of *P. aeruginosa* S3 according to the Victors (89 genes) (Table 3) and VFDB (239 genes) database (S2 Table in S1 File). CARD and NDARO databases were used to find genes responsible for antibiotic resistance of S3 as shown in Table 4. *P. aeruginosa* S3 was found to contain many multidrug efflux systems and also showed the presence of antibiotic resistance genes of fosfomycin, chloramphenicol, and beta-lactam antibiotics. Eight genes across the genome were predicted through the PATRIC database to be related to hemolysin action displayed by S3 (Table 5). A BLASTP-based study revealed that S3 carried two hemolysin genes PapA and ChoE which were

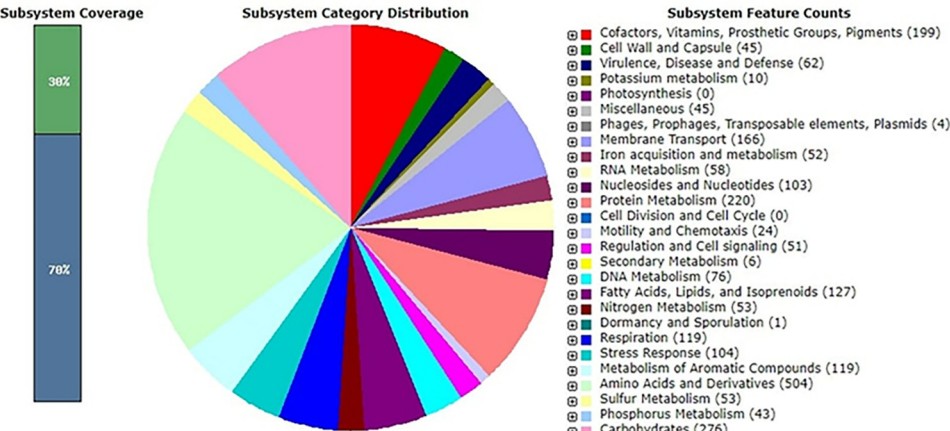

**Fig 5. An annotated draft whole genome of *Pseudomonas aeruginosa* S3 from the RAST server was analyzed to identify subsystem categories.** The pie chart revealed the number of genes related to individual subsystems. Subsystem coverage is presented in the bar graph in the left. The ratio of coding sequences annotated in the SEED subsystem (30%) and outside of the SEED subsystem (70%) is indicated.

annotated as Phospholipase/lecithinase/hemolysin. In addition, apart from the known *plcH* gene, a predicted homolog of the hemolysin III family protein and another putative hemolysin belonging to the GNAT family were present. Rest three genes encoded activation or secretion proteins of hemolysin.

## Discussion

The presence of pathogens in river water causes the spread of several diseases, ultimately with serious consequences for human health [45]. In this present research, a pathogenic strain of *P. aeruginosa* S3 resistant to multiple antibiotics was isolated from the Mahananda river. Hemolytic activity, a key indicator of virulence in several bacteria in the aquatic environment [46]

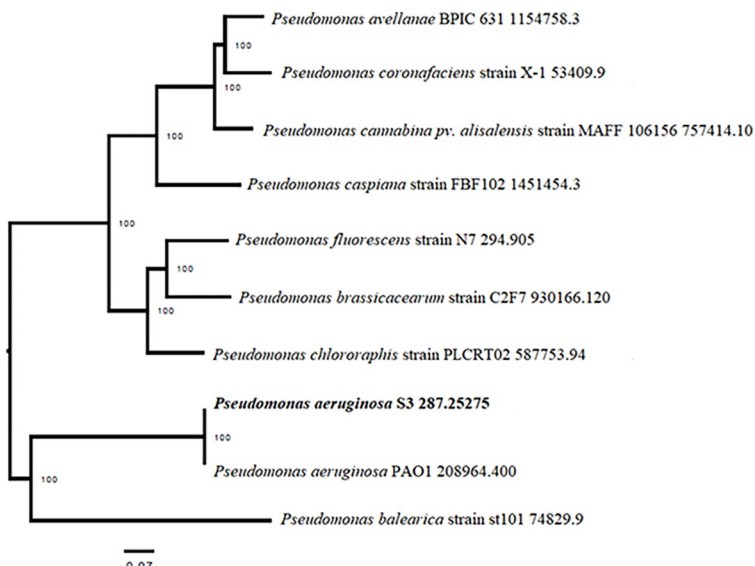

**Fig 6. Phylogenetic analysis based on core genes of *Pseudomonas aeruginosa* S3 using PATRIC.**

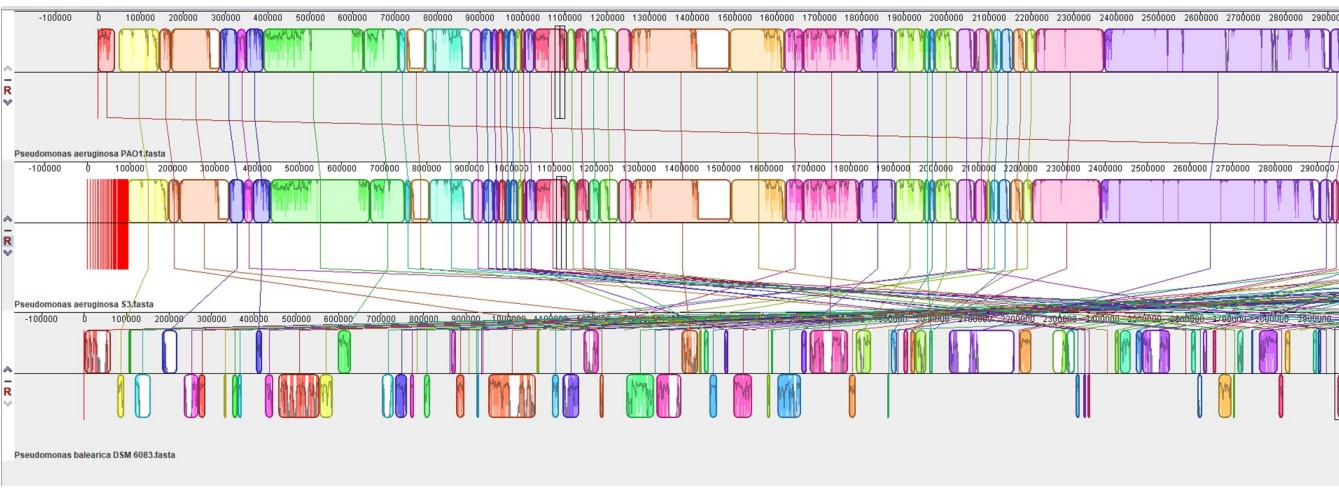

**Fig 7. Genome comparison for *Pseudomonas* strains using Mauve.** Pairwise alignment of *P. aeruginosa* S3 with other whole genomes of *P. aeruginosa* PAO1 and *P. balearica* DSM 6083.

was utilized during the initial screening. Among 26 hemolytic isolates, S3 possessed the highest hemolytic activity and exhibited multiple virulence traits which led to its selection for further studies to genotypically and phenotypically characterize this strain to understand the threat it poses to aquatic organisms and human health.

Virulence factors of *P. aeruginosa* exhibit a wide range of mechanisms controlled by diverse signaling systems and multiple complex and interconnected regulatory circuits, giving this pathogen extraordinary versatility in disease manifestation [10]. Gene prediction in the RAST server showed the existence of Listeria Pathogenicity Island LIPI-1 extended in the S3 genome. All genes of the *L. monocytogenes* LIPI-1 were present in LIPI-1 extended of S3 except *hly* which encodes listeriolysin O [47]. LIPI-1 has been reported to occur in other genera such as in *Bacillus* sp. 87 isolated from food [48]. It was also found in *P. aeruginosa* MZ4A isolated from clinical waste [49]. The presence of LIPI-1 in *P. aeruginosa* S3 might increase its virulence but further studies are required to confirm this possibility.

*P. aeruginosa* is considered one of the significant recurrent emerging pathogens isolated from fishes such as *Oreochromis niloticus*, *Clarias gariepinus* [6,50], *O. mossambicus* [51] and *Labeo bata* [52]. S3 displayed high mortality in fish injection studies with clinical observations agreeing with some previous findings [6,50]. However, as reported by Haque et al. (2021) [52], deep necrotic skin ulcers were not found. In addition, S3 showed cytotoxic activity toward human liver cell lines WRL-68. Previous studies reported *P. aeruginosa* clinical isolates to exhibit cytotoxic properties in human liver cell lines [53] but such studies involving environmental isolates are rare.

Among the various virulence factors found in S3, the two-component system regulatory network comprising signal-sensing histidine kinases and response regulators that play a vital function in *P. aeruginosa* virulence and environmental adaptation [54] were found. *P. aeruginosa* S3 carried the *PA1157* gene which is a two-component response regulator that regulates the response to changes in different environmental conditions. It is a part of the OmpR family which controls porin expression [55]. In addition, the transcriptional regulator gene *algR*, (alginate biosynthesis) which is considered essential for pathogenesis [56] was also detected along with *algB* (response regulator) and *algZ* (sensor histidine kinase). Moreover, the Chp/FimS/AlgR network involved in biofilm formation and virulence was found [57]. In addition, genes encoding the virulence regulator MvfR (PqsR) and the quorum sensing regulator LasR

**Table 3. Prediction of genes related to virulence factors of *Pseudomonas aeruginosa* S3 according to Victors database.**

| Gene | Product | Source ID | Source Organism |
|---|---|---|---|
| nuoD | NADH-ubiquinone oxidoreductase chain C (EC 1.6.5.3) / NADH-ubiquinone oxidoreductase chain D (EC 1.6.5.3) | 15597835 | P.aeruginosa PAO1 |
| PA3001 | NADP-dependent glyceraldehyde-3-phosphate dehydrogenase (EC 1.2.1.13) | 15598197 | P.aeruginosa PAO1 |
| phzS | FAD-dependent monooxygenase PhzS | 15599413 | P.aeruginosa |
| PA14_37260 | Outer membrane low permeability porin, OprD family = > OccK3/OpdO pyroglutamate, cefotaxime uptake | 116050053 | P.aeruginosa UCBPP-PA14 |
| exoT | hypothetical protein | 15595242 | P.aeruginosa PAO1 |
| PA0151 | Ferrichrome-iron receptor @ Iron siderophore receptor protein | 15595349 | P.aeruginosa PAO1 |
| pyrF | Orotidine 5'-phosphate decarboxylase (EC 4.1.1.23) | 15598072 | P.aeruginosa PAO1 |
| algW | Outer membrane stress sensor protease DegS | 15599642 | P.aeruginosa PAO1 |
| prpL | Lysyl endopeptidase (EC 3.4.21.50) | 116052217 | P.aeruginosa UCBPP-PA14 |
| exoY | Adenylate cyclase ExoY (EC 4.6.1.1) | 15597387 | P.aeruginosa PAO1 |
| PA5327 | Oxidoreductase, FAD-binding | 15600520 | P.aeruginosa |
| exoS | putative exoenzyme T | 15599036 | P.aeruginosa PAO1 |
| fur | Ferric uptake regulation protein FUR | 15599958 | P.aeruginosa PAO1 |
| wspF | Chemotaxis response regulator protein-glutamate methylesteraseCheB (EC 3.1.1.61) | 116051700 | P.aeruginosa UCBPP-PA14 |
| htpG | Chaperone protein HtpG | 15596793 | P.aeruginosa PAO1 |
| plcH | Phospholipase C (EC 3.1.4.3) = >hemolyticPlcH | 116048767 | P.aeruginosa UCBPP-PA14 |
| dsbA | Periplasmic thiol:disulfide interchange protein DsbA | 116053639 | P.aeruginosa UCBPP-PA14 |
| PA1157 | Two-component transcriptional response regulator, OmpR family | 15596354 | P.aeruginosa PAO1 |
| estA | Phospholipase/lecithinase/hemolysin | 15600305 | P.aeruginosa PAO1 |
| thrC | Threonine synthase (EC 4.2.3.1) | 15598930 | P.aeruginosa PAO1 |
| pscJ | Type III secretion bridge between inner and outermembrane lipoprotein (YscJ, HrcJ, EscJ, PscJ) | 116049673 | P.aeruginosa UCBPP-PA14 |
| aprF | Type I secretion outer membrane protein, TolC family @ ABC-type protease exporter, outer membrane component PrtF/AprF | 15596445 | P.aeruginosa PAO1 |
| PA3756 | L,D-transpeptidase | 15598951 | P.aeruginosa PAO1 |
| PA14_03530 | Transcriptional regulator | 116053999 | P.aeruginosa UCBPP-PA14 |
| hfq | RNA-binding protein Hfq | 15600137 | P.aeruginosa PAO1 |
| purH | IMP cyclohydrolase (EC 3.5.4.10) / Phosphoribosylaminoimidazolecarboxamideformyltransferase (EC 2.1.2.3) | 15600047 | P.aeruginosa PAO1 |
| pgk | Phosphoglycerate kinase (EC 2.7.2.3) | 15595749 | P.aeruginosa |
| PA0158 | Multidrug efflux system, inner membrane proton/drug antiporter (RND type) = >TriC of TriABC-OpmH system | 15595356 | P.aeruginosa PAO1 |
| PA4489 | UPF0192 protein YfaS | 15599685 | P.aeruginosa |
| relA | Inactive (p)ppGpp 3'-pyrophosphohydrolase domain / GTP pyrophosphokinase (EC 2.7.6.5), (p)ppGpp synthetase I | 15596131 | P.aeruginosa PAO1 |
| wbpX | Mannosyltransferase | 15600642 | P.aeruginosa PAO1 |
| nqrB | Na(+)-translocating NADH-quinone reductase subunit B (EC 1.6.5.8) | 15598194 | P.aeruginosa PAO1 |
| purL | Phosphoribosylformylglycinamidine synthase, synthetase subunit (EC 6.3.5.3) / Phosphoribosylformylglycinamidine synthase, glutamine amidotransferase subunit (EC 6.3.5.3) | 15598958 | P.aeruginosa PAO1 |
| PA14_37650 | hypothetical protein | 116050024 | P.aeruginosa UCBPP-PA14 |
| lasR | N-(3-oxododecanoyl)-L-homoserine lactone-binding transcriptional activator @ Acyl-homoserine lactone-binding transcriptional activator, LuxR family @ Transcriptional regulator LasR | 116049375 | P.aeruginosa UCBPP-PA14 |
| katA | Catalase KatE (EC 1.11.1.6) | 116052272 | P.aeruginosa UCBPP-PA14 |
| narK2 | Nitrate/nitrite transporter NarK/U | 15599071 | P.aeruginosa PAO1 |
| PA3826 | FIG018175: Predicted transmembrane protein | 15599021 | P.aeruginosa |

(*Continued*)

**Table 3.** (*Continued*)

| Gene | Product | Source ID | Source Organism |
|---|---|---|---|
| *PA4115* | Pyrimidine/purine nucleotide 5'-monophosphate nucleosidase PpnN (EC 3.2.2.4) (EC 3.2.2.10) | 15599310 | *P.aeruginosa* |
| *modA* | Molybdenum ABC transporter, substrate-binding protein ModA | 15597060 | *P.aeruginosa* PAO1 |
| *clpV1* | T6SS AAA+ chaperone ClpV (TssH) | 15595288 | *P.aeruginosa* PAO1 |
| *pvdS* | Sigma factor PvdS, controlingpyoverdin biosynthesis | 15597622 | *P.aeruginosa* PAO1 |
| *PA3498* | Flavodoxin reductases (ferredoxin-NADPH reductases) family 1; Vanillate O-demethylase oxidoreductase (EC 1.14.13.-) | 15598694 | *P.aeruginosa* PAO1 |
| *PA0082* | T6SS component TssA (ImpA) | 15595280 | *P.aeruginosa* PAO1 |
| *PA4564* | Conserved uncharacterized protein CreA | 15599760 | *P.aeruginosa* |
| *purD* | Phosphoribosylamine—glycine ligase (EC 6.3.4.13) | 15600048 | *P.aeruginosa* PAO1 |
| *cca* | CCA tRNA nucleotidyltransferase (EC 2.7.7.72) | 15595781 | *P.aeruginosa* PAO1 |
| *mdoG* | Glucans biosynthesis protein G precursor | 15600271 | *P.aeruginosa* PAO1 |
| *fabF1* | 3-oxoacyl-[acyl-carrier-protein] synthase, KASII (EC 2.3.1.179) | 116050967 | *P.aeruginosa* UCBPP-PA14 |
| *flhB* | Flagellar biosynthesis protein FlhB | 116049394 | *P.aeruginosa* UCBPP-PA14 |
| *wzz* | hypothetical protein | 15598356 | *P.aeruginosa* PAO1 |
| *PA4692* | Protein-methionine-sulfoxide reductase catalytic subunit MsrP | 15599886 | *P.aeruginosa* |
| *PA14_04850* | hypothetical protein | 116054099 | *P.aeruginosa* UCBPP-PA14 |
| *galU* | UTP—glucose-1-phosphate uridylyltransferase (EC 2.7.7.9) | 15597219 | *P.aeruginosa* PAO1 |
| *pilI* | type IV pili signal transduction protein PilI | 116054141 | *P.aeruginosa* UCBPP-PA14 |
| *fliC* | Flagellin protein FlaB | 15596289 | *P.aeruginosa* PAO1 |
| *phzM* | Phenazine-specific methyltransferase PhzM | 15599404 | *P.aeruginosa* |
| *PA14_38610* | Short-chain fatty acids transporter | 116049953 | *P.aeruginosa* UCBPP-PA14 |
| *PA2895* | hypothetical protein | 15598091 | *P.aeruginosa* PAO1 |
| *pepA* | Cytosol aminopeptidase PepA (EC 3.4.11.1) | 15599026 | *P.aeruginosa* |
| *pilF* | Type IV pilus biogenesis protein PilF | 116051830 | *P.aeruginosa* |
| *PA0041* | Putative large exoprotein involved in heme utilization or adhesion of ShlA/HecA/FhaA family | 15595239 | *P.aeruginosa* PAO1 |
| *pgm* | 2,3-bisphosphoglycerate-independent phosphoglycerate mutase (EC 5.4.2.12) | 15600324 | *P.aeruginosa* |
| *mvfR* | Multiple virulence factor regulator MvfR/PqsR | 116048931 | *P.aeruginosa* UCBPP-PA14 |
| *PA4887* | Uncharacterized MFS-type transporter | 15600080 | *P.aeruginosa* |
| *PA1009* | Glycine cleavage system transcriptional antiactivatorGcvR | 15596206 | *P.aeruginosa* PAO1 |
| *algU* | RNA polymerase sigma factor RpoE | 15595959 | *P.aeruginosa* PAO1 |
| *PA2972* | Maf-like protein YceF | 15598168 | *P.aeruginosa* PAO1 |
| *PA5312* | Aldehyde dehydrogenase (EC 1.2.1.3) | 15600505 | *P.aeruginosa* |
| *PA14_03120* | Transcriptional regulator, MarR family | 116053966 | *P.aeruginosa* UCBPP-PA14 |
| *pilD* | Leader peptidase (Prepilin peptidase) (EC 3.4.23.43) / N-methyltransferase (EC 2.1.1.-) | 15599724 | *P.aeruginosa* PAO1 |
| *PA4491* | Uncharacterized protein YfaA | 15599687 | *P.aeruginosa* |
| *PA5441* | hypothetical protein | 15600634 | *P.aeruginosa* |
| *PA5437* | Transcriptional regulator PA5437, LysR family | 15600630 | *P.aeruginosa* |
| *PA0073* | ABC transporter, ATP-binding protein | 15595271 | *P.aeruginosa* PAO1 |
| *rhlB* | RhlB, TDP-rhamnosyltransferase 1 (EC 2.4.1.-) | 15598674 | *P.aeruginosa* PAO1 |
| *metE* | 5-methyltetrahydropteroyltriglutamate—homocysteine methyltransferase (EC 2.1.1.14) | 15597123 | *P.aeruginosa* PAO1 |
| *gacA* | BarA-associated response regulator UvrY (= GacA = SirA) | 116050573 | *P.aeruginosa* UCBPP-PA14 |

(*Continued*)

**Table 3.** (Continued)

| Gene | Product | Source ID | Source Organism |
|---|---|---|---|
| *pilY1* | Type IV fimbrial biogenesis protein PilY1 | 15599750 | *P.aeruginosa* PAO1 |
| *PA3286* | beta-ketodecanoyl-[acyl-carrier-protein] synthase (EC 2.3.1.207) | 15598482 | *P.aeruginosa* PAO1 |
| *toxA* | hypothetical protein | 15596345 | *P.aeruginosa* PAO1 |
| *PA3173* | Putative short-chain dehydrogenase | 15598369 | *P.aeruginosa* PAO1 |
| *mucC* | Sigma factor RpoE regulatory protein RseC | 15595962 | *P.aeruginosa* PAO1 |
| *PA14_41070* | FIG005121: SAM-dependent methyltransferase (EC 2.1.1.-) | 116049764 | *P.aeruginosa* UCBPP-PA14 |
| *algR* | Alginate biosynthesis two-component system response regulator AlgR | 15600454 | *P.aeruginosa* PAO1 |
| *PA4488* | Uncharacterized protein YfaQ | 15599684 | *P.aeruginosa* |
| *mucA* | Sigma factor RpoE negative regulatory protein RseA | 15595960 | *P.aeruginosa* PAO1 |

which controls the expression of many virulence factors [58] were detected. *PA1157*, *algB*, *algR* and *algZ* were found to match with *P. aeruginosa* PAO1 genes while MvfR and LasR genes matched with *P. aeruginosa* PA14. However, genes encoding RhlR, another major quorum-sensing regulator, was not found. Nevertheless, the presence of other virulence regulatory genes in *P. aeruginosa* S3 appears to be sufficient to express its pathogenic phenotype. Production of siderophores is reported to be essential in the pathogenesis of *P. aeruginosa* [59,60]. *P aeruginosa* S3, which exhibited siderophore production in the phenotype expression experiment was found to carry thirteen *pvd* cluster genes associated with pyoverdin synthesis and its regulation and secretion; and another nine genes involved in pyochelin synthesis and secretion. We found the presence of several other important virulence-associated genes similar to PAO1 that included *Las A* (protease precursor) and the pseudomonas elastase gene *Las B* (extracellular zinc protease) which render the ability to invade tissues and thus cause damage to host cells [61,62].

Several genes encoding multiple secretion systems that are associated with host colonization and virulence were detected, e.g., genes associated with the type III secretion system which are grouped in five operons comprising structural and regulatory genes *viz. pscNOPQRSTU*, *popNpcr1234DR*, *pcrGVHpopBD*, *exsCEBA*, and *exsDpscBCDEFGHIJKL* [63]. Besides, genes of several effectors associated with the type III secretion system such as *exoS*, *exoY*, and *exoT* were found. ExoS supports cell invasion and intracellular persistence through its ADP-ribosyl-transferase activity [64]. However, *exoU*, considered the most virulent of the T3SS effectors and the main driver of the cytotoxic phenotype was not found in *P. aeruginosa* S3. Isolation of virulent *P. aeruginosa* strains lacking one or more such effectors is not unusual and PAO1, known to be hemolytic and cytotoxic [19], also lacks this gene. Moreover, the presence of *exoS* and *exoU* is almost exclusive [63]. In the context that some high-risk strains carry either *exoU* or *exoS*; authors have suggested that environmental strains are more likely to carry *exoS* which is more prevalent while strains of specific clinical origin mostly carry *exoU* [65]. Despite not carrying *exoU*, the cytotoxic phenotype showed by *P. aeruginosa* S3 nevertheless implies that it is a health risk that demands attention. Genes associated with the type IV secretion system which matched with *P. aeruginosa* PAO1 included *pilA* (the major abundant pilin subunit PilA), *fimUpilVWXY1Y2E* operon (several less abundant, fiber-associated pilin-like proteins and surface sensor), the *pilMNOPQ* operon (proteins necessary for assembly and twitching motility), and PilD, the prepilin leader peptidase needed for processing the major and minor pilins [57,66,67] were also found. This system is involved in host tissue adherence and colonization and promoting surface-associated twitching motility [67]. In addition, S3, like PAO1, carried the virulence locus HSI-1 of the type VI secretory system which structurally resembles

**Table 4. Prediction of antimicrobial resistance genes of *Pseudomonas aeruginosa* S3 based on CARD, and NDARO database.**

| Gene | Product | Source ID | Source Organism |
|------|---------|-----------|-----------------|
| *oprN* | Multidrug efflux system, outer membrane factor lipoprotein = >OprN of MexEF-OprN system | NP_251185.1 | *P.aeruginosa* PAO1 |
| *opmE* | Multidrug efflux system, outer membrane factor lipoprotein = >OpmE of MexPQ-OpmE system | BAE06009.1 | *P.aeruginosa* |
| *mexL* | Transcriptional regulator, AcrR family | NP_252368.1 | *P.aeruginosa* PAO1 |
| *PmrB* | Sensory histidine kinase QseC | AEX49906.1 | *P.aeruginosa* |
| *mexM* | RND efflux system, membrane fusion protein | BAE06005.1 | *P.aeruginosa* |
| *oprD* | Outer membrane low permeability porin, OprD family = > OccD1/OprD basic amino acids, carbapenem uptake | NP_249649.1 | *P.aeruginosa* PAO1 |
| *gyrA* | DNA pase subunit A (EC 5.99.1.3) | NP_251858.1 | *P.aeruginosa* PAO1 |
| *amrB* | Multidrug efflux system, inner membrane proton/drug antiporter (RND type) = >MexY of MexXY/AxyXY | NP_250708.1 | *P.aeruginosa* PAO1 |
| *mexE* | Multidrug efflux system, membrane fusion component = >MexE of MexEF-OprN system | NP_251183.1 | *P.aeruginosa* PAO1 |
| *mexH* | Multidrug efflux system, membrane fusion component = >MexH of MexHI-OpmD system | NP_252895.1 | *P.aeruginosa* PAO1 |
| *mexZ* | Transcriptional repressor of mexXY operon, MexZ | NP_250710.1 | *P.aeruginosa* PAO1 |
| *parC* | DNA topoisomerase IV subunit A (EC 5.99.1.3) | BAA37152.1 | *P.aeruginosa* |
| *oprM* | Multidrug efflux system, outer membrane factor lipoprotein = >OprM of MexAB-OprM | NP_249118.1 | *P.aeruginosa* PAO1 |
| *TriC* | Multidrug efflux system, inner membrane proton/drug antiporter (RND type) = >TriC of TriABC-OpmH system | NP_248848.1 | *P.aeruginosa* PAO1 |
| *oprJ* | Multidrug efflux system, outer membrane factor lipoprotein = >OprJ of MexCD-OprJ system | AAB41958.1 | *P.aeruginosa* |
| *TriB* | Multidrug efflux system, membrane fusion component = >TriB of TriABC-OpmH system | NP_248847.1 | *P.aeruginosa* PAO1 |
| *phoQ* | Sensor histidine kinase PhoQ (EC 2.7.13.3) | NP_249871.1 | *P.aeruginosa* PAO1 |
| *nalC* | Transcriptional regulator, AcrR family | NP_252410.1 | *P.aeruginosa* PAO1 |
| *mexD* | Multidrug efflux system, inner membrane proton/drug antiporter (RND type) = >MexD of MexCD-OprJ system | NP_253288.1 | *P.aeruginosa* PAO1 |
| *mexA* | Multidrug efflux system, membrane fusion component = >MexA of MexAB-OprM | NP_249116.1 | *P.aeruginosa* PAO1 |
| *OpmH* | Outer membrane channel TolC (OpmH) | NP_253661.1 | *P.aeruginosa* PAO1 |
| *mexK* | Multidrug efflux system, inner membrane proton/drug antiporter (RND type) = >MexK of MexJK-OprM/OpmH system | AAG07064.1 | *P.aeruginosa* PAO1 |
| *FosA* | Fosfomycin resistance protein FosA | NP_249820.1 | *P.aeruginosa* PAO1 |
| *mexW* | Multidrug efflux system, inner membrane proton/drug antiporter (RND type) = >MexW of MexVW-OprM system | AAG07763.1 | *P.aeruginosa* PAO1 |
| *mexF* | Multidrug efflux system, inner membrane proton/drug antiporter (RND type) = >MexF of MexEF-OprN system | NP_251184.1 | *P.aeruginosa* PAO1 |
| *PmrA* | Two-component system response regulator QseB | NP_253464.1 | *P.aeruginosa* PAO1 |
| *mexN* | Multidrug efflux system MdtABC-TolC, inner-membrane proton/drug antiporter MdtB-like | BAE06006.1 | *P.aeruginosa* |
| *TriA* | Multidrug efflux system, membrane fusion component = >TriA of TriABC-OpmH system | NP_248846.1 | *P.aeruginosa* PAO1 |
| *PDC-1* | Class C beta-lactamase (EC 3.5.2.6) = > PDC family | ACQ82807.1 | *P.aeruginosa* PAO1 |

*(Continued)*

**Table 4.** (Continued)

| Gene | Product | Source ID | Source Organism |
|------|---------|-----------|-----------------|
| *mexV* | Multidrug efflux system, membrane fusion component = >MexV of MexVW-OprM system | AAG07762.1 | *P.aeruginosa* PAO1 |
| *mexC* | Multidrug efflux system, membrane fusion component = >MexC of MexCD-OprJ system | AAB41956.1 | *P.aeruginosa* |
| *mexS* | Putative oxidoreductase | ADT64081.1 | *P.aeruginosa* PAK |
| *mexI* | Multidrug efflux system, inner membrane proton/drug antiporter (RND type) = >MexI of MexHI-OpmD system | NP_252896.1 | *P.aeruginosa* PAO1 |
| *arnA* | UDP-4-amino-4-deoxy-L-arabinose formyltransferase (EC 2.1.2.13) / UDP-glucuronic acid oxidase (UDP-4-keto-hexauronic acid decarboxylating) (EC 1.1.1.305) | NP_252244 | *P.aeruginosa* PAO1 |
| *mexR* | Multidrug resistance operon repressor MexR, MarR family | NP_249115.1 | *P.aeruginosa* PAO1 |
| *parE* | DNA topoisomerase IV subunit B (EC 5.99.1.3) | NP_253654.1 | *P.aeruginosa* PAO1 |
| *mexP* | Multidrug efflux system, membrane fusion component = >MexP of MexPQ-OpmE system | BAE06007.1 | *P.aeruginosa* |
| *mexB* | Multidrug efflux system, inner membrane proton/drug antiporter (RND type) = >MexB of MexAB-OprM | AAA74437.1 | *P.aeruginosa* |
| *nfxB* | Transcriptional regulator NfxB | NP_253290.1 | *P.aeruginosa* PAO1 |
| *amrA* | Multidrug efflux system, membrane fusion component = >MexX of ofMexXY/AxyXY | NP_250709.1 | *P.aeruginosa* PAO1 |
| *catB7* | Chloramphenicol O-acetyltransferase (EC 2.3.1.28) = >CatB family | NP_249397.1 | *P.aeruginosa* PAO1 |
| *mexJ* | Multidrug efflux system, membrane fusion component = >MexJ of MexJK-OprM/OpmH system | NP_252367.1 | *P.aeruginosa* PAO1 |
| *opmD* | Multidrug efflux system, outer membrane factor lipoprotein = >OpmD of MexHI-OpmD system | NP_252897.1 | *P.aeruginosa* PAO1 |
| *nalD* | Transcriptional regulator, AcrR family | NP_252264.1 | *P.aeruginosa* PAO1 |
| *APH(3')-IIb* | Aminoglycoside 3'-phosphotransferase (EC 2.7.1.95) = > APH(3')-II/APH(3')-XV | CAA62365.1 | *P.aeruginosa* |
| *mexG* | hypothetical protein | NP_252894.1 | *P.aeruginosa* PAO1 |
| *OXA-50* | Class D beta-lactamase (EC 3.5.2.6) = > OXA-50 family, oxacillin-hydrolyzing | AAQ76277.1 | *P.aeruginosa* |
| *mexQ* | Multidrug efflux system, inner membrane proton/drug antiporter (RND type) = >MexQ of MexPQ-OpmE system | BAE06008.1 | *P.aeruginosa* |
| *phoP* | Transcriptional regulatory protein PhoP | NP_249870.1 | *P.aeruginosa* PAO1 |
| *emrE* | small multidrug resistance family (SMR) protein | NP_253677.1 | *P.aeruginosa* PAO1 |

the contractile tails of bacteriophages [68]. Hemolysin-coregulated protein (Hcp) and valine–glycine repeat protein G (VgrG) act as major structural components and effectors of this system which are associated with key functions in virulence, biofilm formation and competition with microorganisms in the environment [57,69]. The presence of these versatile secretary machinery might play a role in the pathogenicity of *P. aeruginosa* S3.

Antibiotic resistance machinery in bacteria develops through a spectrum of genomic changes [70]. Genomic studies revealed the presence of two crucial beta-lactamases, class C beta-lactamase *PDC-1* and class D beta-lactamase *OXA-50*, which occur naturally in *P. aeruginosa* [71]. Therefore, *P. aeruginosa* is, in general, resistant to the beta-lactam class of antibiotics [72,73] but, higher susceptibility is often found towards cephalosporins and imipenem in both clinical [74] and environmental [75] isolates. Notably, our strain showed complete resistance towards all beta-lactam class of antibiotics. The expression of class C beta-lactamase PDC

**Table 5. Hemolysin genes prediction of *Pseudomonas aeruginosa* S3 based on PATRIC database.**

| Hemolysin-related gene annotations (PATRIC) | Identity of the Gene (based on NCBI BLASTP) | Percentage identity(%) | Source organism |
| --- | --- | --- | --- |
| Hemolysin activation/secretion protein associated with VreARIsignalling system | ShlB/FhaC/HecB | 100 | *Pseudomonas* sp. |
| Putative hemolysin | GNAT family N(alpha)-acetyltransferase | 100 | *Pseudomonas aeruginosa* |
| Hemolysin activation/secretion protein | ShlB/FhaC/HecB | 100 | *Pseudomonas* sp. |
| Hemolysin activation/secretion protein | two-partner secretion system transporter CdrB | 100 | *Pseudomonas* sp. |
| FIG01964566: Predicted membrane protein, hemolysin III homolog | Hemolysin III family protein | 99.51 | *Pseudomonas aeruginosa* |
| Phospholipase/lecithinase/hemolysin | cholinesterase/ ChoE | 100 | *Pseudomonas* sp. |
| Phospholipase/lecithinase/hemolysin | PapA | 100 | *Pseudomonas aeruginosa* PAO1 |
| Phospholipase C, hemolytic PlcH | phospholipase C, phosphocholine-specific | 100 | *Pseudomonas* sp. |

(Pseudomonas-derived cephalosporinase) is considered the main resistance mechanism in *P. aeruginosa* against the beta-lactam class of antibiotics [76]. On the other hand, the most common type of carbapenemases is the Class D β-lactamases, also known as oxacillinases (OXA) which hydrolyses imipenem [71]. The presence of both *PDC-1* and *OXA-50* in the draft genome of S3 might play a role in the resistance shown by S3 towards all beta-lactam type of antibiotics. However, other mechanisms might also be involved, including those of RND (resistance-nodulation-cell division) family efflux pumps. Apart from the intrinsic mechanisms, several efflux systems have been characterized in *P. aeruginosa* which contribute more actively to resistance by extrusion of antibiotics from the cell [77]. Genes encoding ten of these multidrug efflux systems belonging to the RND family which are mexAB-OprM, mexC-D-OprJ, mexEF-OprN, mexGHI-OpmD, mexJK-OprM, mexMN-OprM, mexPQ-OmpE, mexVW-OprM, mexXY-OprM and TriABC-OmpH were found in S3. In addition, the gene encoding efflux pump of the SMR (small multidrug resistance) family, namely *emrE* was also detected [78]. All of these genes matched with PAO1. While the expression of most of these is tightly regulated, overexpression resulting from antibiotic exposure occurs due to mutations in regulatory proteins [77]. Resistance to tetracyclines, folate pathway inhibitors and norfloxacin in S3 observed in this study might be due to overexpression of some of the efflux systems or mutations in the topoisomerase genes *gyrA/gyrB* and *parA/parB* [79,80]. However, further studies are required to explain these resistant phenotypes. Antibacterial colistin (polymyxin E) has recently received increasing attention as a last-resort treatment option against multidrug-resistant Gram-negative bacteria [81]. Reported data suggests that adaptive resistance to polymyxins in Gram-negative bacteria occurs due to modifications of outer membrane lipopolysaccharide (LPS) mediated by *pmrHFIJKLME* operon which is under the regulation of the two two-component systems, PhoPQ and PmrAB [82]. Equally important is resistance mediated by the *mexAB-oprM* operon, which is involved in nonspecific antimicrobial efflux [83]. The *pmrHFIJKLME* operon was not detected in S3 but the *mexAB-oprM* operon and the genes coding for regulatory components PmrAB and PhoPQ were found. Colistin resistance in S3 may be due to other mechanisms preliminarily confirmed to occur in colistin-resistant *P. aeruginosa* [84,85]. Hence, further studies on molecular mechanisms involved in colistin resistance in S3 should be of interest. Aminoglycoside resistance in *P. aeruginosa* is mainly due to the structural modification by three types of enzymes, aminoglycoside phosphotransferase (APH), aminoglycoside acetyltransferase (AAC), and aminoglycoside nucleotidyl transferase (ANT). APH in *P. aeruginosa* transfers a phosphoryl group to the 3′-hydroxyl of aminoglycosides, making antibiotics such as streptomycin and kanamycin inactive [86]. The *APH* gene was

found in S3 and matched with PAO1, but the other two genes were absent which is consistent with the finding that S3 displayed resistance to kanamycin and streptomycin but not to genta-mycin. The chloramphenicol resistance gene *catB7* encoding chloramphenicol O-acetyltrans-ferase was detected which corroborates with chloramphenicol resistance observed in the disc diffusion test but identification of the exact mode of resistance, especially in the presence of the robust efflux system which can use chloramphenicol as substrate requires further investiga-tions. Additionally, *fosA* encoding the fosfomycin resistance protein, widely distributed among several Gram-negative bacteria including PAO1 [87] was found.

Five genes were predicted as putative hemolysins in *P. aeruginosa* S3. It included the *plcH* gene that encodes hemolytic phospholipase C [4]. This gene, found to be upregulated in bio-film-forming cells compared to planktonic cells [88], was also reported to be present in hemo-lytic *P. aeruginosa* strains isolated from environmental sources [89]. The capability of *P. aeruginosa* to form biofilms is associated with its ability to induce long-lasting chronic lung infections and antibiotic resistance [90]. Phospholipase C contributes to the breakdown of lip-ids and lecithin and facilitates its establishment in the respiratory tract. Moreover, it helps in its survival within the tissue by suppressing the respiratory burst response of neutrophils [91]. Another gene predicted as hemolysin was cholinesterase (ChoE) which is reported to act together with PlcH and PchP (acid phosphatase) in *P. aeruginosa* to obtain choline and inor-ganic phosphate as nutrients for bacterial metabolism [92]. However, PchP was not detected in the draft genome of S3 thus suggesting the occurrence of multiple modes of action. It was observed that *P. aeruginosa* cholinesterase (PA4291 gene) hydrolyses acetylcholine and is regarded as a pathogenic factor that supports corneal infection [93]. Clinical changes in the fish eyes induced by S3 were also observed in the current study but understanding the extent of involvement of this gene here requires further studies. Another predicted gene was identi-fied as the hemolysin III family protein. Well-characterized in *B. cereus*, hemolysin III family proteins are regarded as pore-forming toxins [94]. But, the role of this gene in pathogenicity in *P. aeruginosa* remains to be studied. The fourth putative hemolysin was a GNAT family pro-tein known as a large group of enzymes that acylates various substrates using acyl-coenzyme A [95]. They execute a wide range of cellular functions, including the activation of pore-forming toxins by a distinct group of this family known as toxin-activating acyltransferase [96]. Shin and Choe [97] characterized the PA4534 protein from *P. aeruginosa*, a GNAT protein, and found it closely related to N-terminal acetyltransferase. The fifth gene predicted to encode a putative hemolysin in S3 was identified as the PapA gene which codes for type P pilin in *E. coli* and is involved in uropathogenicity [98]. All of these genes matched with several distinct strains of *Pseudomonas*, including PAO1. The participation of all or some of these proteins might be instrumental in the strong hemolytic activity observed in *P. aeruginosa* S3 and further research on understanding their contribution towards hemolysin action should be of interest.

## Conclusion

This study reported the genotypic and phenotypic characterization of *P. aeruginosa* S3, a path-ogenic and antibiotic-resistant strain isolated from the Mahananda River. The draft genome of S3 provided better insights into the virulent and antibiotic-resistant phenotype. The presence of eight genes related to hemolysin action was detected. In addition, several genes involved in antibiotic efflux mechanisms were found. To our knowledge, this is the first report characteriz-ing pathogenic and antibiotic-resistant *P. aeruginosa* from this river though, additional research is required to comprehend how the virulent and antibiotic resistance genes function fully. *P. aeruginosa* is a common recurrent pathogen frequently isolated from clinical settings. However, the isolation of a virulent and antibiotic-resistant strain from river water raises

public health concerns and gives a warning about the appropriate use of antibiotics. Therefore, periodic monitoring of *P. aeruginosa* in the river water is needed as it indicates poor water quality that could lead to disease outbreaks.

## Supporting information

**S1 File.**
(DOCX)

## Acknowledgments

The authors would like to thank the Department of Biotechnology and Department of Botany, University of North Bengal for helping in carrying out the entire research work.

## Author Contributions

**Conceptualization:** Dipanwita Ghosh, Dipanwita Saha.

**Data curation:** Dipanwita Ghosh, Anoop Kumar, Protip Basu.

**Formal analysis:** Dipanwita Ghosh, Preeti Mangar, Anoop Kumar, Aniruddha Saha, Protip Basu, Dipanwita Saha.

**Investigation:** Dipanwita Ghosh, Abhinandan Choudhury.

**Methodology:** Dipanwita Ghosh, Abhinandan Choudhury.

**Project administration:** Dipanwita Saha.

**Resources:** Aniruddha Saha, Dipanwita Saha.

**Software:** Dipanwita Ghosh, Protip Basu.

**Supervision:** Dipanwita Saha.

**Validation:** Dipanwita Saha.

**Visualization:** Dipanwita Ghosh, Preeti Mangar, Protip Basu.

**Writing – original draft:** Dipanwita Ghosh.

**Writing – review & editing:** Dipanwita Ghosh, Dipanwita Saha.

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
