## [Decision Letter · Decision Letter 0]

18 Dec 2023

PONE-D-23-33885Characterization of a hemolytic and antibiotic-resistant Pseudomonas aeruginosa strain S3 pathogenic to fish isolated from Mahananda River in IndiaPLOS ONE

Dear Dr. Saha,

Thank you for submitting your manuscript to PLOS ONE. After careful consideration, we feel that it has merit but does not fully meet PLOS ONE’s publication criteria as it currently stands. Therefore, we invite you to submit a revised version of the manuscript that addresses the points raised during the review process.

We look forward to receiving your revised manuscript.

Kind regards,

Seyed Mostafa Hosseini

Academic Editor

PLOS ONE

Journal Requirements:

2. Please upload a new copy of Figure 4 as the detail is not clear. Please follow the link for more information: " ext-link-type="uri" xlink:type="simple">https://blogs.plos.org/plos/2019/06/looking-good-tips-for-creating-your-plos-figures-graphics/"
" ext-link-type="uri" xlink:type="simple">https://blogs.plos.org/plos/2019/06/looking-good-tips-for-creating-your-plos-figures-graphics/"

Reviewers' comments:

Reviewer's Responses to Questions

**Comments to the Author**

1. Is the manuscript technically sound, and do the data support the conclusions?

Reviewer #1: Yes

Reviewer #2: Yes

Reviewer #3: Yes

2. Has the statistical analysis been performed appropriately and rigorously? 

Reviewer #1: Yes

Reviewer #2: No

Reviewer #3: Yes

3. Have the authors made all data underlying the findings in their manuscript fully available?

Reviewer #1: Yes

Reviewer #2: Yes

Reviewer #3: Yes

4. Is the manuscript presented in an intelligible fashion and written in standard English?

Reviewer #1: No

Reviewer #2: Yes

Reviewer #3: Yes

5. Review Comments to the Author

Reviewer #1: 1- The text of the article should be revised in terms of grammar.

2- A number of words in the pdf file are stuck together, to be checked and corrected. Words that are stuck together should be separated.

3- The author claimed that the strain isolated from the river water is very similar to the standard PAO1 strain, can it be said that there is a similarity in terms of other characteristics such as the ability of hemolysis, antibiotic resistance, etc.? Explain and compare.

4- Explain the reasons for conducting cytotoxicity test for the identified strain?

5- What method was used to investigate the resistance of the isolated strain S3 to the colistin? Disc diffusion method?

Reviewer #2: I have no comments

Reviewer #3: 1. It is not interesting to write S3 at the beginning of the abstract, it is better to give the first set of bacteria names (Virulent strain Pseudomonas aeruginosa) .

2. In the abstract, the names of the genes should be written in italics

3. What was the reason for not using CLSI as a reference for the antibiogram?

4. "Some authors" in reference number 8 is not interesting, it is better to write "some researchers"

5. Afterincubation should be corrected in the materials and methods section (section Hemolytic activity).

6. It would have been better to determine the typing sequence of the strain using the MLST technique

6. PLOS authors have the option to publish the peer review history of their article (what does this mean?). If published, this will include your full peer review and any attached files.

Reviewer #1: No

Reviewer #2: No

Reviewer #3: No

---

## [Author Response · Author response to Decision Letter 0]

3 Feb 2024

Author Response- The suggested revision has been done.

.

2. Please upload a new copy of Figure 4 as the detail is not clear. Please follow the link for more information: https://blogs.plos.org/plos/2019/06/looking-good-tips-for-creating-your-plos-figures-graphics/" https://blogs.plos.org/plos/2019/06/looking-good-tips-for-creating-your-plos-figures-graphics/"

Author Response- A new copy of Figure 4 has been uploaded.

Author Response-The suggested revision has been done.

Reviewers' comments:

Reviewer's Responses to Questions

Comments to the Author

1. Is the manuscript technically sound, and do the data support the conclusions?

Reviewer #1: Yes

Reviewer #2: Yes

Reviewer #3: Yes

2. Has the statistical analysis been performed appropriately and rigorously?

Reviewer #1: Yes

Reviewer #2: No

Reviewer #3: Yes

3. Have the authors made all data underlying the findings in their manuscript fully available?

Reviewer #1: Yes

Reviewer #2: Yes

Reviewer #3: Yes

4. Is the manuscript presented in an intelligible fashion and written in standard English?

Reviewer #1: No

Reviewer #2: Yes

Reviewer #3: Yes

5. Review Comments to the Author

Reviewer #1: 1- The text of the article should be revised in terms of grammar.

Author Response 1- The text has been revised for grammar

2- A number of words in the pdf file are stuck together, to be checked and corrected. Words that are stuck together should be separated.

Author Response 2- The stuck words were separated wherever found. 

3- The author claimed that the strain isolated from the river water is very similar to the standard PAO1 strain, can it be said that there is a similarity in terms of other characteristics such as the ability of hemolysis, antibiotic resistance, etc.? Explain and compare.

Author Response 3- Following the suggestion, some comparisons are now included in the manuscript in the discussion section. These regions are highlighted in the manuscript.

4- Explain the reasons for conducting cytotoxicity test for the identified strain?

Author Response 4- Cytotoxicity test was done to comprehend the spectrum of virulence properties carried by S3.

5- What method was used to investigate the resistance of the isolated strain S3 to the colistin? Disc diffusion method?

Author Response 5- Yes disc diffusion method was used to investigate the resistance of the isolated strain S3 to colistin.

However, as rightly questioned by the reviewer, three tests are accepted for colistin sensitivity test in the CLSI guidelines which includes broth micro-dilution, disk elution and colistin agar method. Thus, the disk diffusion method is not acceptable. To overcome this shortfall, we performed the disk elution method and the result showed that S3 was resistant to colistin. This information is now included in the manuscript in materials methods section. 

Reviewer #2: I have no comments

Reviewer #3: 1. It is not interesting to write S3 at the beginning of the abstract, it is better to give the first set of bacteria names (Virulent strain Pseudomonas aeruginosa).

Author Response 1-The suggested revision has been done.

2. In the abstract, the names of the genes should be written in italics

Author Response 2-The suggested revision has been done.

3. What was the reason for not using CLSI as a reference for the antibiogram?

Author Response 3-We have followed both the EUCAST and CLSI guidelines but inadvertently, CLSI was not mentioned in the manuscript. This is now included.

4. "Some authors" in reference number 8 is not interesting, it is better to write "some researchers"

Author Response 4-The suggested revision has been done.

5. Afterincubation should be corrected in the materials and methods section (section Hemolytic activity).

Author Response 5-The suggested revision has been done.

6. It would have been better to determine the typing sequence of the strain using the MLST technique

Author Response 6-The sequence type was found as 549 using the MLST database which also matched with P. aeruginosa PAO1. This information is already included in the manuscript; hence no changes were done.

6. PLOS authors have the option to publish the peer review history of their article (what does this mean?). If published, this will include your full peer review and any attached files.

Do you want your identity to be public for this peer review? For information about this choice, including consent withdrawal, please see our Privacy Policy.

Reviewer #1: No

Reviewer #2: No

Reviewer #3: No

---

## [Decision Letter · Decision Letter 1]

22 Feb 2024

Characterization of a hemolytic and antibiotic-resistant Pseudomonas aeruginosa strain S3 pathogenic to fish isolated from Mahananda River in India

PONE-D-23-33885R1

Dear Dr. Dipanwita Saha

We’re pleased to inform you that your manuscript has been judged scientifically suitable for publication and will be formally accepted for publication once it meets all outstanding technical requirements.

Kind regards,

Seyed Mostafa Hosseini

Academic Editor

PLOS ONE

Additional Editor Comments (optional):

Reviewers' comments:

Reviewer's Responses to Questions

**Comments to the Author**

1. If the authors have adequately addressed your comments raised in a previous round of review and you feel that this manuscript is now acceptable for publication, you may indicate that here to bypass the “Comments to the Author” section, enter your conflict of interest statement in the “Confidential to Editor” section, and submit your "Accept" recommendation.

Reviewer #1: All comments have been addressed

Reviewer #3: All comments have been addressed

2. Is the manuscript technically sound, and do the data support the conclusions?

Reviewer #1: Yes

Reviewer #3: Yes

3. Has the statistical analysis been performed appropriately and rigorously? 

Reviewer #1: Yes

Reviewer #3: Yes

4. Have the authors made all data underlying the findings in their manuscript fully available?

Reviewer #1: Yes

Reviewer #3: Yes

5. Is the manuscript presented in an intelligible fashion and written in standard English?

Reviewer #1: Yes

Reviewer #3: (No Response)

6. Review Comments to the Author

Reviewer #1: (No Response)

Reviewer #3: (No Response)

7. PLOS authors have the option to publish the peer review history of their article (what does this mean?). If published, this will include your full peer review and any attached files.

Reviewer #1: No

Reviewer #3: No

---

## [Editor Report · Acceptance letter]

20 Mar 2024

PONE-D-23-33885R1 

PLOS ONE

Dear Dr. Saha, 

I'm pleased to inform you that your manuscript has been deemed suitable for publication in PLOS ONE. Congratulations! Your manuscript is now being handed over to our production team.

Kind regards, 

on behalf of

Dr. Seyed Mostafa Hosseini 

Academic Editor

PLOS ONE